# Human Motion Tracking Using 3D Image Features with a Long Short-Term Memory Mechanism Model—An Example of Forward Reaching

**DOI:** 10.3390/s22010292

**Published:** 2021-12-31

**Authors:** Kai-Yu Chen, Li-Wei Chou, Hui-Min Lee, Shuenn-Tsong Young, Cheng-Hung Lin, Yi-Shu Zhou, Shih-Tsang Tang, Ying-Hui Lai

**Affiliations:** 1Department of Biomedical Engineering, National Yang Ming Chiao Tung University, Taipei 112, Taiwan; s930470.be09@nycu.edu.tw (K.-Y.C.); leo641001@gmail.com (Y.-S.Z.); 2Department of Physical Therapy and Assistive Technology, National Yang Ming Chiao Tung University, Taipei 112, Taiwan; lwchou@nycu.edu.tw; 3The Research Center on ICF and Assistive Technology, National Yang Ming Chiao Tung University, Taipei 112, Taiwan; noralee0724@gmail.com; 4Institute of Geriatric Welfare Technology & Science, MacKay Medical College, New Taipei City 252, Taiwan; styoung@mmc.edu.tw; 5Department of Electrical Engineering, National Taiwan Normal University, Taipei 106, Taiwan; brucelin@ntnu.edu.tw; 6Department of Biomedical Engineering, Ming Chuan University, Taoyuan 333, Taiwan; sttang@mail.mcu.edu.tw; 7Medical Device Innovation & Translation Center, National Yang Ming Chiao Tung University, Taipei 112, Taiwan

**Keywords:** depth image, time-of-flight camera, deep learning, human motion tracking, rehabilitation application

## Abstract

Human motion tracking is widely applied to rehabilitation tasks, and inertial measurement unit (IMU) sensors are a well-known approach for recording motion behavior. IMU sensors can provide accurate information regarding three-dimensional (3D) human motion. However, IMU sensors must be attached to the body, which can be inconvenient or uncomfortable for users. To alleviate this issue, a visual-based tracking system from two-dimensional (2D) RGB images has been studied extensively in recent years and proven to have a suitable performance for human motion tracking. However, the 2D image system has its limitations. Specifically, human motion consists of spatial changes, and the 3D motion features predicted from the 2D images have limitations. In this study, we propose a deep learning (DL) human motion tracking technology using 3D image features with a deep bidirectional long short-term memory (DBLSTM) mechanism model. The experimental results show that, compared with the traditional 2D image system, the proposed system provides improved human motion tracking ability with RMSE in acceleration less than 0.5 (m/s^2^) X, Y, and Z directions. These findings suggest that the proposed model is a viable approach for future human motion tracking applications.

## 1. Introduction

Rehabilitation is becoming an increasingly important issue owing to the rise in elderly population. According to the World Population Ageing 2020 report [1], 727 million people were aged 65 years or older around the world, and the number of the elderly was projected to rise to 1.5 billion by 2050. In addition, the World Health Organization (WHO) indicates that 15 million people suffer from stroke each year [2], with 75% of those being elderly [3]. This means that healthcare, such as motion rehabilitation to regain motor function, must be valued and improved.

Rehabilitation is described as a set of interventions that assist individuals with health concerns in achieving and maintaining optimal functioning in their daily lives [4]. Motor rehabilitation requires accurate measurement of limb movement for evaluation and treatment purposes. However, according to the statistics from the WHO, there are less than 10 skilled practitioners per one million people worldwide, which means a shortage of physical therapists [5]. To improve motor rehabilitation quality, many types of motion capture (MoCap) systems have been developed to monitor human movement more efficiently.

There are three main types of MoCap systems: non-visual-based tracking, visual marker-based tracking, and visual marker-free tracking [3]. In non-visual-based tracking systems, inertial measurement unit (IMU) devices are broadly used to obtain motion information based on inertia-based microelectromechanical system sensors that contain accelerators, gyroscopes, and magnetometers. IMU devices are cost-efficient and easy to use, and human motion tracking tasks can be performed with these devices thanks to their high sensitivity and accuracy. For example, Seel et al. used IMU sensors to analyze gait performance and accurately determined the root mean square errors (RMSE) of the knee flexion/extension angles, with less than 1° differences on a prosthesis and 3° on human legs compared with a visual marker-based tracking system [6]. Similarly, this application performed well in an upper limb analysis. Jung et al. measured human arm movements using inertial sensors, and the results indicated that the displacement errors of the three axes, that is, the coordinates of the arm segments, were less than 5 cm [7]. These studies demonstrate the applicability of IMU sensors for human motion tracking. However, IMU sensors are limited as they must be attached to the body, which can be inconvenient or cause discomfort [8]. Therefore, visual marker-based tracking systems such as Vicon [9] and visual marker-free tracking systems can be used to alleviate these issues [10,11,12,13].

Currently, Vicon is a well-known visual marker-based tracking system that contains several infrared-light cameras. Cameras track the markers that are attached to human skin from different points of view. The system then calculates the precise spatial positions of each marker from the recordings. Previous studies proved that the Vicon system could provide high accuracy for motion tracking [14,15]. However, this system is difficult to popularize and apply clinically because of its high cost and low portability. Conversely, the visual marker-free tracking system is another well-known approach that is mainly based on image processing technology and computer vision. Recently, OpenPose has become a well-known system for human motion tracking tasks [10]. This system estimates human poses based on RGB images and predicts locations of human joints and the affinity between two joints using convolution neural network (CNN) technology [16]. OpenPose has been applied to human motion analysis and rehabilitation tasks, and the results have demonstrated good performance [17,18,19]. However, these applications also reveal the disadvantages of using RGB images for human motion tracking. As the performance of motion estimation relies on the lighting conditions [17] and recording directions [18], the environmental setup must be properly controlled for target tasks. More specifically, precise 3D information cannot be accurately obtained because single RGB camera systems have difficulty capturing depth features. These types of systems generally have limited recording views, which means that similar motions will represent different features in the 2D motion capture system. As a result, planar motions are easier to analyze, whereas non-planar motions are difficult to accurately detect with a 2D image system. Therefore, a multiple-RGB-camera system is used to solve this accuracy problem. For example, Li et al. used a two-camera system to analyze human gaits, and the results showed that the accuracy could be improved compared to a single camera system [20]. Although the two-camera system provides better accuracy, it is not convenient enough for clinical users because the cameras have to be calibrated. Additional problems exist for RGB-based systems, such as ambient lighting conditions, which directly affect the accuracy of visual recognition [21].

More recently, single RGB-depth (RGB-D) camera systems, known as time-of-flight (ToF) cameras, have been applied to human motion tracking [22]. The ToF camera sensor can obtain depth information through its emitter and receiver. The emitter discharges infrared light on the object, whereas the receiver obtains the reflected light and calculates the depth information through the time difference. In addition, lower cost and higher portability make it popular for research and even for daily use. Based on the advantages of the RGB-D sensors, Chen et al. used CNN technology to perform regression analysis of the IMU information, demonstrating that RGB-D sensor technologies not only improve the performance of human motion tracking but also provide more useful information of 3D features than of 2D features [23]. Moreover, Ma et al. used long short-term memory (LSTM) [24], a deep learning (DL) method proficient in temporal information processing, to rectify the kinematic information obtained from the ToF camera [25]. Specifically, the *x*-, *y*-, and *z*-axis acceleration information originating from human motion were captured and used as the regression goal. The results indicated that it is feasible to use the DL method to study the accuracy of human motion tracking systems. At the same time, using an LSTM-type model has advantages over temporal information processing. Based on the successful results noted above, the purpose of this study is to prove whether a deep bidirectional LSTM (DBLSTM) mechanism model [26] with 3D image features can further improve the performance of human motion tracking. Conventional models, including a deep neural network (DNN) [27] and CNN, were compared to investigate the benefits of temporal information in human motion tracking tasks. Note that the “arm reaching” task, an important training task in rehabilitation, was chosen as the test example in this study, as it is a common daily function for the upper extremities. In addition, the features of classic RGB images, known as 2D images, were used as the baseline system for comparison to investigate the benefits of the proposed system compared to 3D images.

Our main contributions of this study are as follows: (1) We investigate whether the time-related deep learning model structure, DBLSTM, is advantageous compared to several traditional models in human motion tracking tasks. Employing IMU sensors as our target provided *x*-, *y*-, and *z*-axis acceleration information, which are a kind of continuous signal and closely related to time clues. We took our system as a kind of regression approach to obtain the acceleration information from 3D image features, and the results demonstrated that DBLSTM indeed performed well in such kinds of tasks. (2) We delve into the potentiality of 3D image features to determine whether the data comprise more information that can be representative of human motion. The 3D image comprises features such as the relative relationship—i.e., involving vectors and Euclidean distances—between the human joints. Two-dimensional image features were used as a comparison. Data from several recording views were collected and evaluated through our proposed system, and the results did verify the high potentiality for 3D images.

## 2. The Proposed System

A block diagram of the proposed system, including the training and testing phases, is shown in Figure 1. In the training phase, the amplitude ($M_{A}$) and point cloud information ($M_{P}$) were recorded using a ToF camera. Specifically, $M_{A}$ refers to the energy of the images, which are 2D grayscale images; $M_{P}$ refers to the point cloud data, which are generated through the information of $M_{A}$ and phases that are obtained through the receiver on the ToF camera. Representations of $M_{A}$ and $M_{P}$ are shown in Figure 2. Next, $M_{A}$ was input to the OpenPose system to obtain the human joints ($J_{2D}$), which were represented by two-dimensional coordinates. Images were input into the OpenPose system, and human poses were estimated using the CNN. Confidence maps and part affinity fields were set up to detect human body parts and part associations from different people. Part affinity was first calculated through CNN to analyze the vector information of human poses. For example, based on the colors of the images, the vectors offered information linking the shoulder to the elbow, the elbow to the wrist, and so on. Subsequently, the obtained information was combined with the original input, that is, means the image, and sent to the next convolutional layers to analyze the positions of the human joints [10]. All of the joints from the bottom to the top were obtained through OpenPose. Based on our previous study [23], four human joints were selected for the “arm reaching” action: the right shoulder (RS), left shoulder (LS), left elbow (LE), and left wrist (LW). The LS point was chosen as the reference point. Note that these four points were the most efficient positions in the observation of “arm reaching” actions in our previous study.

The features for the human joints in the 2D images ($J_{2D}$) can be expressed as:(1)$$J_{2D}=\left\{\begin{array}{c} RS_{2D}\left(x,y\right)-LS_{2D}\left(x,y\right),\\ LE_{2D}\left(x,y\right)-LS_{2D}\left(x,y\right),\\ LW_{2D}\left(x,y\right)-LS_{2D}\left(x,y\right),\\ D_{2D} \end{array}\right\},$$
where $RS_{2D}\left(x,y\right)$, $LS_{2D}\left(x,y\right)$, $LE_{2D}\left(x,y\right)$, and $LW_{2D}\left(x,y\right)$ are the joint coordinates of the right shoulder, left shoulder, left elbow, and left wrist, respectively; $D_{2D}$ represents the Euclidean distances from the right shoulder, left elbow, and left wrist joints to the left shoulder joint. More specifically, $D_{2D}$ can be expressed as:(2)$$D_{2D}=\left\{\begin{array}{c} \|RS_{2D}\left(x,y\right)-LS_{2D}\left(x,y)\right\|,\\ \|LE_{2D}\left(x,y\right)-LS_{2D}\left(x,y)\right\|,\\ \|LW_{2D}\left(x,y\right)-LS_{2D}\left(x,y)\right\| \end{array}\right\}.$$

A 3D generator unit was used to combine the information of $J_{2D}$ and $M_{P}$ [23]; $J_{2D}$ assists the generator to determine the depth coordinates of the images from the distribution of the point cloud data, which is denoted by $J_{3D}$. The same feature selection as for the 2D images was adapted, with the joint of the left shoulder used as the reference point to obtain relative vectors and distances. In this case, the features for the human joints in the 3D images ($J_{3D}$) can be expressed as:(3)$$J_{3D}=\left\{\begin{array}{c} RS_{3D}\left(x,y,z\right)-LS_{3D}\left(x,y,z\right),\\ LE_{3D}\left(x,y,z\right)-LS_{3D}\left(x,y,z\right),\\ LW_{3D}\left(x,y,z\right)-LS_{3D}\left(x,y,z\right),\\ D_{3D} \end{array}\right\},$$
where $RS_{3D}\left(x,y,z\right)$, $LS_{3D}\left(x,y,z\right)$, $LE_{3D}\left(x,y,z\right)$, and $LW_{3D}\left(x,y,z\right)$ represent the joint coordinates for the right shoulder, left shoulder, left elbow, and left wrist, respectively; $D_{3D}$ represents the Euclidean distances from the right shoulder, left elbow, and left wrist to the left shoulder. More specifically, $D_{3D}$ can be expressed as:(4)$$D_{3D}=\left\{\begin{array}{c} \|RS_{3D}\left(x,y,z\right)-LS_{3D}\left(x,y,z)\right\|,\\ \|LE_{3D}\left(x,y,z\right)-LS_{3D}\left(x,y,z)\right\|,\\ \|LW_{3D}\left(x,y,z\right)-LS_{3D}\left(x,y,z)\right\| \end{array}\right\}.$$

In the training phase, the $J_{2D}$ and $J_{3D}$ data were separately used as inputs for the 2D and 3D regression model tasks, respectively. The same target (G) was used for both to evaluate performance. The obtained $J_{3D}$ was used as the input for the proposed DBLSTM model, which has the advantage of processing temporal information such as natural language and speech recognition tasks [26,28]. Therefore, it was assumed that the DBLSTM model would provide further benefits for tracking tasks because human motion is a type of spatial movement and continuous signal. The synchronously recorded data G from the IMU sensor was used as the output of the DBLSTM model. The G signals included the *x*-, *y*-, and *z*-axis acceleration information, which can be expressed as:(5)$$G=\left\{Acc_{x},Acc_{y},Acc_{z}\right\},$$
where $Acc_{x}$, $Acc_{y}$, and $Acc_{z}$ are the accelerations of the *x*-, *y*-, and *z*-axis, respectively, obtained from the IMU sensors. The mean squared error was used as the loss function to ensure convergence of the model and evaluate the predicted results. The related equation can be expressed as:(6)$$W^{*},B^{*}=\arg\underset{W,B}{\min}\sum_{i=1}^{t}{\left(G_{i}-{\hat{G}}_{i}\right)}^{2},$$
where t indicates the total frame of the training data; $G_{i}$ and ${\hat{G}}_{i}$ are the training target and predicted result in the ith time frame, respectively; $W^{*}$ and $B^{*}$ represent the optimal weight and bias parameters of the training model, respectively. Finally, the ToF camera was used to record the human motion and subsequently obtain the predicted motion information in the testing phase.

## 3. Materials and Methods

### 3.1. Materials

The data from the IMU sensors and the ToF camera were recorded at 50 fps and were temporally aligned. “Arm reaching” [29] was used as the target task because it is the most commonly executed motor function by the upper extremities and a major focus of stroke rehabilitation. The setup of the recording environment is shown in Figure 3. During the experiment, the IMU sensors were placed on the subject’s left wrist, and the ToF camera was placed in front of the subject. The subject was seated behind a cup with their arm resting on the table. During the repetition of the task, the subject was asked to perform the arm reaching task repetitively. The whole sequences of “arm reaching” included: (1) the subject extending the left arm and using the left hand to pick up and raise the cup to the height that matched their mouth, and (2) the subject returning the cup to the starting point. As the experiment started, the subject would first raise the hand on the table for approximately 5 s; then, the “arm reaching” motion started. Until the time was up, the subject had to lower their hand. An Xsens IMU sensor [30] and RL100 Pro camera were used [31]. The IMU sensor was placed flat on the table before the experiment started to ensure that the calibrated *x*-, *y*-, and *z*-axis directions were the same in every recording. Considering the inherent deficit of cumulative error from the accelerometers, they were calibrated before the experiments were held. The data had been analyzed to compensate for accelerator errors, resulting in the minimization of cumulative errors. Data were recorded for a total of 1.5 h in three types of views ($S_{1}$, $S_{2}$, and $S_{3}$, as shown in Figure 3). The positions of $S_{1}$ and $S_{3}$ represented the subject sitting at relative angles of 0° (frontal view) and 30°, respectively. These two recording views were used as training data (or defined as known-view data), whereas the $S_{2}$ recording view (sitting at a relative angle of 15°) was used as the testing data (or defined as unknown-view data). Note that the known-view data means the DL model recognizes the recording view of the data, and the unknown-view data means the DL model does not recognize the recording view of the data. Three different recording views were used to simulate the real conditions.

### 3.2. Methods

Two experiments were conducted: the first was for the comparison of model performance, and the second was for the comparison between the performances of the 2D and 3D systems. The purpose of the first experiment was to address the question of whether the DBLSTM model could demonstrate better results in the human motion regression task. In the second experiment, the DBLSTM model was further used to compare the performance of 2D and 3D image features. We recorded several types of data acquired from different recording views, as shown in Figure 3, but the same type of motion to simulate the real conditions for daily life and rehabilitation use. Considering the conditions for the users, the recording views in every system operation may not be sufficiently similar. Consequently, the system should tolerate some displacement resulting from the recording view. Therefore, the second experiment was conducted to analyze whether 2D or 3D image features can be applied to the usage of different recording views, leading to a more robust system.

In the first experiment, the DNN and CNN models were compared with the DBLSTM model. The DNN structure is the most commonly used model in machine learning and includes several hidden layers consisting of multilayer perceptrons [27] that calculate the relationship between the input and output using back propagation technology [32]. A DNN was selected as our baseline owing to its simple structure. CNN is a well-known DL structure frequently used in image recognition and signal analysis [33], which obtains spatial or temporal information through convolutional calculations. In our previous study, a CNN was used to regress the information of IMU signals from 3D images, and it was proven to provide suitable performance [23]. Here, the known-view data were used to evaluate the performance of the system. The data were split into training and testing sets at a ratio of 9:1. The setting information of the three models is listed in Table 1. In addition, batch normalization [34] was used in each hidden layer, and the Nadam optimizer [35] was used to optimize the models. The initial learning rate was set to 0.0005, and the batch size was set to 1024.

In the second experiment, the performance between the 2D and 3D features was determined. The DBLSTM model, which had the best performance (see details in Section 4.1) in the first experiment, was also used in this experiment. The recording views from $S_{1}$ and $S_{3}$ were used to train the DL model. Two parts were set up to evaluate the robustness of the performance under different conditions, defined as the known-view and unknown-view performances. The known-view performance used all the recording views of the testing data that were recognized in the system, whereas the unknown-view performance used distinct recording views of the testing data that were not recognized in the system. The same training data were used as in the first experiment, which included recording views from $S_{1}$ and $S_{3}$. In addition, the recording view from $S_{2}$ was used to evaluate the unknown-view performance.

### 3.3. Evaluation

The RMSE [36], percent root mean square error (PRMSE) [37], and coefficient of multiple correlations (CMC) [38] were used to determine the performance of the proposed system. Each axis was separated to evaluate the performance. The RMSE was used to indicate the differences between the labeled data and predicted results. The PRMSE, which is a method based on the RMSE, provided a more accurate index, which was determined using the mean of the data. The PRMSE has the advantage of demonstrating better precision because of the variances of the three axes. The PRMSE equation is expressed as:(7)$$\mathrm{PRMSE}=\sqrt{\frac{\sum_{i=1}^{t}\frac{{\left(G_{i}-{\hat{G}}_{i}\right)}^{2}}{G_{i}}}{t}}\times 100\%,$$
where $G_{i}$ and ${\hat{G}}_{i}$ represent the testing target and predicted result in the ith time frame, respectively. In this type of method, it helps to evaluate the percentage of the error that is produced by the system.

The CMC indicates whether the relationship between $G_{i}$ and ${\hat{G}}_{i}$ is well connected and is expressed as:(8)$$\mathrm{CMC}=1-\sum_{i=1}^{t}\frac{{\left(G_{i}-{\hat{G}}_{i}\right)}^{2}}{{\left(G_{i}-\bar{G}\right)}^{2}},$$
where $\bar{G}$ is the average of the test target. The evaluation of the CMC can be found in [39]. The CMC value was limited to 0−1, where CMC ≈ 1 indicates an excellent relationship between the testing target and predicted results, 0.5 < CMC < 0.99 indicates a good relationship, 0.16 < CMC < 0.49 indicates a moderate connection, and CMC ≤ 0.16 indicates a poor relationship. The predicted results provided three types of information: the *x*-, *y*-, and *z*-axis accelerations, which were evaluated separately. Here, lower RMSE and PRMSE values with a higher CMC value indicate that the system exhibited better performance.

## 4. Results and Discussion

### 4.1. Model Performance Comparison

The results of the model performance comparison, with the best performances marked in bold, are listed in Table 2. First, all the models demonstrated good performance with CMC > 0.9 and PRMSE < 5%, which means that using DL methods can obtain good performance in human motion tracking tasks. Moreover, the DBLSTM model achieved the best CMC performance with the least RMSE and PRMSE among these models. Specifically, the DBLSTM model predicted the most accurate results compared with the labeled data. CNN was ranked second, and DNN performed the worst. The performance might depend on the characteristics of each model. The DBLSTM model holds an advantage when processing temporal information. No studies have compared the performance for the three types of DL methods simultaneously in human motion tracking tasks. Several studies in the field of speech recognition and human activity recognition showed that DBLSTM exhibited a more stable and accurate performance compared with such kinds of traditional DL models [28,39]. The proposed DBLSTM model considered the IMU acceleration as the target of the regression model and proved that the method was feasible. Even though the concept of the DBLSTM is not that novel, it is a state-of-the-art model in terms of analyzing temporal information. DBLSTM comprises LSTM units, which are characteristic of its forget gate, input gate, and output gate. The forget gate includes a sigmoid function to manipulate the information originating from the previous unit, whereas the input and output gates include sigmoid and hyperbolic tangent functions to reinforce the important information originating from its unit mixed with the message coming from the forget gate. The most important information can be extracted and passed to the next LSTM unit. Based on the message that human motion is a continuous signal, adequate parameters manipulated by time sequence can make the model perform better, and the increased hidden layers help the model be more robust. Moreover, CNN can capture subsequent information through kernels and convolution. The same features were used to demonstrate a similar performance, but the performance shows the limits. DNN owns the fundamental concept in deep learning, and it does exhibit the basic performance. It is clear that acceleration information is highly related to the temporal changes. In conclusion, the DBLSTM model succeeded in extracting useful information, obtaining more accurate results than the CNN and DNN models.

### 4.2. Comparison of 2D/3D Performance

Based on the results of the first experiment, we selected DBLSTM to evaluate the performance of the 2D and 3D experiments. The results of the 2D and 3D performance experiments are shown in Figure 4. The horizontal axis represents the number of time frames, whereas the vertical axis shows the value of the acceleration data. The results of the *x*-, *y*-, and *z*-axis accelerations from known-view data are shown in Figure 4a–c, respectively. The results of the *x*-, *y*-, and *z*-axis accelerations from the unknown-view data are shown in Figure 4d–f, respectively. There are three lines in each subplot: blue represents the testing target from the IMU sensors, yellow represents the predicted results from 2D image features, and orange represents the predicted results from the 3D image features. The 3D image features performed slightly better than the 2D image features, as shown in Figure 4a–c. More detailed information about the known-view and unknown-view performances are listed in Table 3 and Table 4, respectively. Using 3D features to regress human motion information was confirmed to be more accurate than using 2D features, as shown in Table 4. Both of the features in the known-view conditions achieved good performance with CMC > 0.9 and PRMSE < 5%. We think that the DBLSTM model was successful in extracting the features of human motion based on its memory cell structure. In addition, the amount of motion data was sufficient for the system to handle this task. Our results demonstrate that using 2D images could achieve a similar accuracy level as 3D images from the known recording views of the data. However, for the unknown recording views of the data, the 2D image had increased prediction errors, whereas the 3D image maintained high accuracy, as shown in Figure 4d–f.

The detailed performance results are presented in Table 4. The 2D features clearly performed well with increased RMSE and PRMSE, and a low CMC. Conversely, the 3D features achieved good performance with CMC > 0.9 and PRMSE < 10%. The difference between the 2D and 3D image features was the depth information, which can be referred to as the *z*-axis information in $J_{3D}$. The depth information assisted the system to obtain spatial messages when subjects were performing motions, thus improving the accuracy. In addition, every kind of recording view was different for the 2D images because the features lacked depth information. Even though a subject performed the same type of motion, it appeared to be different for 2D image features because of the different recording views. The 2D system performed well under the known-view condition but was challenged under the unknown-view condition. This implies that 3D image features have an advantage over 2D image features in unknown-view applications. The CMC in the unknown-view performance was as good as in the known-view performance. However, the PRMSE in the unknown-view performance appeared slightly higher than that in the known-view performance. The linear relationship between the predicted results and the IMU data was good, and the PRMSE appeared to be slightly higher than expected. It is assumed that the errors could be decreased by increasing the recording views for the provided human motion data.

## 5. Conclusions

This study proposed a DL-based system to track human motion using 3D image features with a DBLSTM model. Three-dimensional image features were combined with the DL model to improve the accuracy of human motion tracking and increase the tolerance of the recording view. The “arm reaching” movement was used as a target task to evaluate the performance, which showed that such features could succeed in regressing motion information. A depth camera was used to capture the motion of the subject, providing adequate information for doctors or therapists to evaluate important features of the patients’ performance. The main contributions of this study are as follows.

(1) Classic DL models were used to compare the performance of a human motion regression task. The results demonstrated that the DBLSTM model had an advantage over DNN and CNN models because of its ability to retain temporal information. Because the acceleration information from IMU sensors are continuous signals, models that can process time sequences are advantageous. As we can see, the DBLSTM was demonstrated to predict the most accurate results based on similar parameters.

(2) The model’s predictability with 2D and 3D image features was compared, and the data from multiple recording views were used to confirm the advantages of the DBLSTM model. The results showed that the 3D system demonstrated superiority in recording the spatial information of human motion. This is because 3D image features contained depth information that was able to capture spatial changes. The application of 3D image features was first discovered to demonstrate the potential for unknown-view applications. The relative relationships between chosen human joints were the features used. The information includes the relative vectors and Euclidean distances. In our opinion, the features help build a robust connection between joints that carry out the same motion type from different recording views. The relative information from different recording views seems to be stable in 3D image features. As we can see, the left elbow moves relative to the left wrist with the same motions, which can be defined as a rigid relationship. The unknown recording view $S_{2}$ is likely to refer to the interpolation of the perspective of known views $S_{1}$ and $S_{3}$, and it shows that applications for unknown views are likely to succeed. However, we cannot guarantee whether other recording views, especially referring to extrapolations in 45°, 60°, or higher, can achieve the same performance as in the interpolated case.

In summary, the proposed system has the potential for clinical application. Looking ahead to the application of tele-rehabilitation, there are increasingly more opportunities to use cameras to track and evaluate the condition of patients. The proposed system provided good information that indicated the acceleration of human motion. However, considering future applications in clinical use, more data from different recording views may be needed. The proposed system demonstrated potential for use in tele-rehabilitation, but its practicality must be verified through subsequent studies. We also recognize that the calibration process only reduces but does not eliminate the motion integration error based on the acceleration retrieved from the accelerometer. In practical applications, it is still necessary to analyze the error function that changes over time, so that the solution we propose can perform better. In future applications that use camera sensors with depth photography capabilities, patients will not need to wear any sensors to record the signals, and body movement could be tracked with a single camera, resulting in a convenient and practical way to improve the delivery of the clinical application.

## Figures and Tables

**Figure 1 sensors-22-00292-f001:**
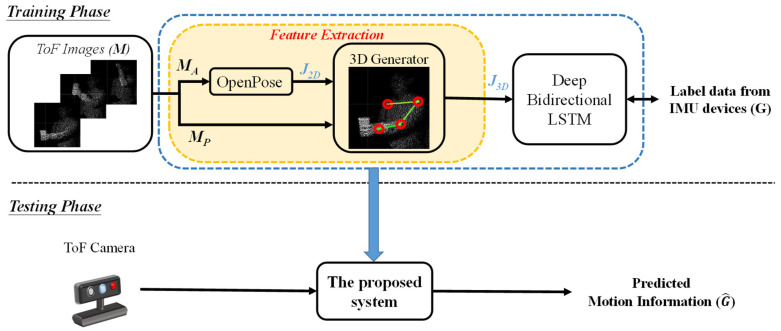
Structure of the proposed system.

**Figure 2 sensors-22-00292-f002:**
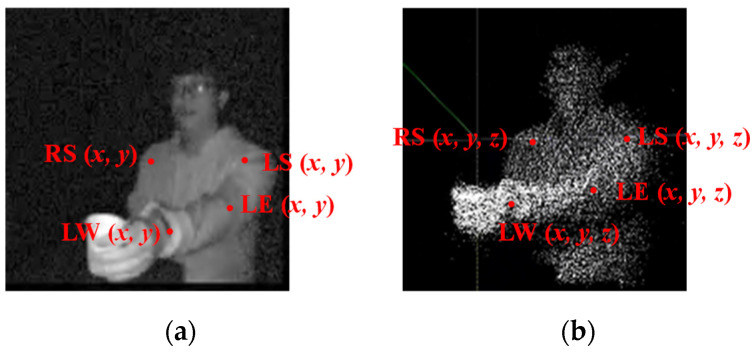
Concepts for: (**a**) 2D images ($M_{A}$), which comprise gray-scale images, and (**b**) 3D images ($M_{P}$ ), which consist of a point cloud.

**Figure 3 sensors-22-00292-f003:**
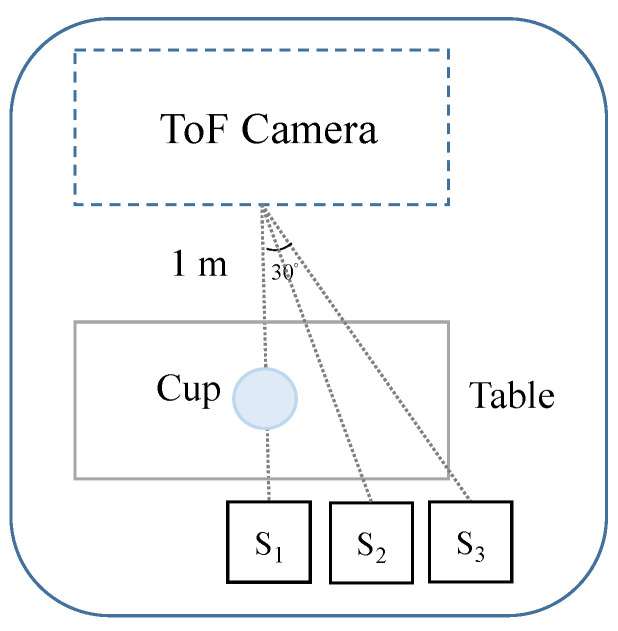
Setup of the recording environment: $S_{1}$ and $S_{3}$ stand for the known-view data, whereas $S_{2}$ stands for the unknown-view data.

**Figure 4 sensors-22-00292-f004:**
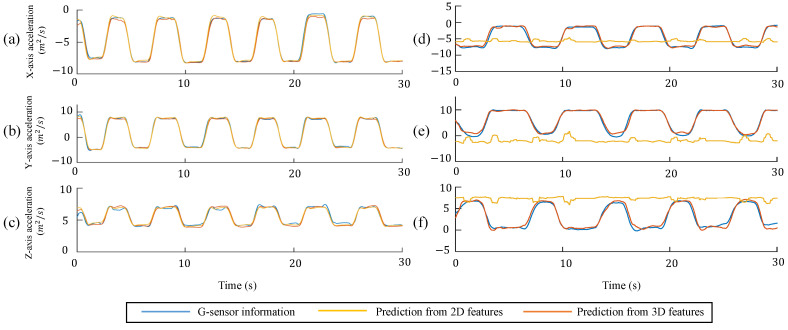
2D/3D performance comparison; (**a**–**c**) represent the results of the *x*-, *y*-, and *z*-axis accelerations from known-view data, respectively; (**d**–**f**) represent the results of the *x*-, *y*-, and *z*-axis accelerations from unknown-view data, respectively.

**Table 1 sensors-22-00292-t001:** Details for model setting.

Model	Hidden Layer	Total Parameters
DNN	Neurons = (36, 324, 324, 324, 36) Dropout = (None, 0.3, 0.2, None, None)	501,988
CNN	Filters = (32, 64, 128, 256) Kernel size = 3 × 3 Dropout = (None, 0.3, 0.2, None)	512,644
DBLSTM	Bidirectional LSTM, units = 128 LSTM, units = 256 Dropout = (0.3, 0.2)	500,996

**Table 2 sensors-22-00292-t002:** Details for model performance comparison. Note that the best performances are marked in bold.

	DNN	CNN	DBLSTM
***X*-axis acceleration**			
**RMSE**	0.30	0.26	**0.22**
**PRMSE**	3.2%	2.8%	**2.3%**
**CMC**	0.990	0.993	**0.995**
***Y*-axis acceleration**			
**RMSE**	0.48	0.40	**0.29**
**PRMSE**	4.3%	3.6%	**3.5%**
**CMC**	0.991	**0.994**	**0.994**
***Z*-axis acceleration**			
**RMSE**	0.38	0.37	**0.33**
**PRMSE**	3.4%	3.3%	**3.0%**
**CMC**	0.926	0.930	**0.943**

**Table 3 sensors-22-00292-t003:** Details for 2D/3D performance comparison in known-view testing. Note that the best performances are marked in bold.

	3D Images	2D Images
***X*-axis acceleration**		
**RMSE**	**0.22**	0.25
**PRMSE**	**2.3%**	2.6%
**CMC**	**0.995**	0.993
***Y*-axis acceleration**		
**RMSE**	**0.29**	0.47
**PRMSE**	**3.5%**	4.3%
**CMC**	**0.994**	0.991
***Z*-axis acceleration**		
**RMSE**	**0.33**	0.34
**PRMSE**	**3.0%**	**3.0%**
**CMC**	**0.943**	0.941

**Table 4 sensors-22-00292-t004:** Details for 2D/3D performance comparison in unknown-view testing. Note that the best performances are marked in bold.

	3D Images	2D Images
***X*-axis acceleration**		
**RMSE**	**0.37**	2.92
**PRMSE**	**4.7%**	37.2%
**CMC**	**0.984**	0.004
***Y*-axis acceleration**		
**RMSE**	**0.46**	4.02
**PRMSE**	**2.7%**	23.5%
**CMC**	**0.987**	0.010
***Z*-axis acceleration**		
**RMSE**	**0.50**	5.88
**PRMSE**	**8.5%**	42.5%
**CMC**	**0.960**	0.007

## Data Availability

The data that support the findings of this study are available at https://reurl.cc/zWGzV7 (accessed on 1 December 2021).

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
