# Peer review of "Human Motion Tracking Using 3D Image Features with a Long Short-Term Memory Mechanism Model—An Example of Forward Reaching"

_sensors, 2021, doi:10.3390/s22010292_

Round 1

Reviewer 1 Report

The following are comments and suggestions for possible publicatio of this paper.

1) Main contributions should be in the introduction section.

2) Title of section 3 should be Materials and Methods.

3) Explanation about methods should be under the subsection of Method.

 4) More experiments should be performed using the different action data.

Author Response

Dear reviewer, thank you for your great comments. For the detailed response please see the attachment. Thank you very much. 

Reviewer 2 Report

The paper introduces a deep learning method for human motion tracking using 3D image features with a DBLSTM model. The paper is very well written. The method used is simple, straightforward.

I do have several comments:

Major:

1) In my opinion, the idea of using IMU acceleration as the target of the model needs to be discussed. The raw IMU data is not gold-standard. IMU data includes bias which causes problems in the estimation of the position (final target). In normal cases, we use the Vicon system as reference data to evaluate the position extracted from the IMU-acceleration data.

Hence, I do not get the main idea of using DL to estimate acceleration data from 2D and 3D images. Sure, you have demonstrated the ability to do that with DL, but please explain the relevance of this task. Since you do not have any information on the sensor bias, you can not process these data further to the final target (position).

Please explain the relevance of your work regarding this point?

2) For quality control of the data and the synchronization you used, please provide an example of a data set that includes IMU data, as well as calculated 2D and 3D parameters over time.

Minor:

1) Figure 4: please change the x-axis unit into seconds or add the sampling rate to the caption.

2) Figure 4: y-axis label for d) e) f) missing.

Author Response

(The authors gave the same response as above.)

Reviewer 3 Report

The authors have proposed a manuscript describing RGB-D (ToF camera) based tracking of the upper extremity. They exploited well established Bi-LSTM deep model supported with 3D coordinates relations. Unfortunately, the proposed contributions seem not spectacular, and the research methodology reveals noticeable flaws.

The state-of-the-art section is not representative since the Bi-LSTM deep model is widely studied in the literature for limb motion tracking, and its possible novelty is unclear. Adequate additional references should be supplemented and critically commented as to emphasise authors’ contribution in this field. There is a shortage of recent achievements in the research domain. Comparison with DNN and CNN approaches seem to be not challenging nowadays. Especially the results do not seem to be outstanding – exemplary CMC metrics varies in an insignificant place.

Moreover, both the proposed system as well as the method reveal considerable flaws and imprecisions. Exemplary what do amplitude (Ma) and point cloud information (Mp) mean – there is no adequate explanation? As the introduced system strictly relates to the OpenPose system, it should be sufficiently described, despite being a well-known solution. In consequence structure of the proposed method is not clear and should be revised. The idea of the superiority of “3D image features” is abstract. Authors treat the composition of 3D joints coordinates as a feature vector and confront it with a feature vector composed of 2D joints’ in image positions. The superiority of the 3D approach is well studied and almost trivial. The statement that the “DBLSTM model is successful in extracting the features of human motion” requires at least a deeper explanation.

Considering the experiment scenario, it must be noted that “arm reaching”, though intuitively clear, formally remained a highly undefined movement, especially in the context of deep model output parameters. It may vary among different users and even among subsequent gesture repetitions – this aspect was not discussed. Additionally, acceleration output issues can not be treated as reliable motion tracking determinants as IMU-based acceleration deficits are well-known – i.e. cumulative error, etc. This aspect was not also discussed. As the angle S_2 implied interpolation of recorded parameters from the perspective of known angles S_1 and S_3 related parameters, it is difficult to determine what is the generalisation potential of the deep model and how it behaves in the case of unknown views – interpolation issues are usually simpler than extrapolation issues. In consequence, the method “increase in tolerance of the recording view” was not supported by the experiments.

Summing up, the manuscript should be considerably supplemented and revised before reconsideration for publication. In the present form, it requires at least major revision.

Author Response

(The authors gave the same response as above.)

Round 2

Reviewer 2 Report

The authors have carefully revised the manuscript.

Reviewer 3 Report

The supplementations and clarifications, provided by the authors, are certainly valuable and should be appreciated. Nevertheless, the main concerns, regarding the authors' contributions in the area under consideration, unfortunately, remained.

It is incomprehensible to make hypotheses contesting the superiority of 3D features over 2D features in the research domain – “main purpose of the experiment is to see if the 3D image features own more efficient information than 2D image features did”, as this has been demonstrated in many previous works. The quantitative difference is obvious, and its value depends on the movements, the accuracy of the devices and the time over which the tracking takes place if cyclic calibration activities are not carried out.

2D features created from the RGB-D signal involve an obvious loss of information and comparing them to the effectiveness of 3D features is inappropriate. The authors should confront other tracking methods using 3d features.

The calibration process only reduces but does not eliminate motion integration errors based on acceleration retrieved from an accelerometer. The error function over time has not been analysed, which limits the functionality of the proposed solution.

Summing up the manuscript should be thoroughly revised due to the above deficits. Special care should be put to quality metrics: “performance”, “robustness”, “proficient” which were not supported by the provided results.
